# Genotype by Environment Interactions (G*E) of Chickens Tested in Ethiopia Using Body Weight as a Performance Trait

**DOI:** 10.3390/ani13193121

**Published:** 2023-10-06

**Authors:** Maud A. J. de Kinderen, Johann Sölkner, Gábor Mészáros, Setegn W. Alemu, Wondmeneh Esatu, John W. M. Bastiaansen, Hans Komen, Tadelle Dessie

**Affiliations:** 1Division of Livestock Science, University of Natural Resources and Life Sciences, Gregor-Mendel-Straße 33, 1180 Vienna, Austria; 2Department of Animal Breeding and Genomics, Wageningen University and Research, P.O. Box 338, 6700 AH Wageningen, The Netherlands; 3International Livestock Research Institute, Addis Ababa P.O. Box 5689, Ethiopia

**Keywords:** Ethiopia, smallholder farming, poultry, body weight, agro-ecologies, G*E

## Abstract

**Simple Summary:**

Smallholder poultry farming plays a major socio-economic role in Ethiopia and other African developing nations by securing food and income for local households. One way of trying to optimize smallholder poultry production is testing introduced commercial dual-purpose strains into new African environments. Different strains may be best suited for different environments of Ethiopia, which is a country with a wide range of agro-ecologies. A traditional way of investigating performance differences between strains and their new environments is statistical testing for genotype by environment interaction (G*E). Our G*E results suggest that body weight performance varies between chicken strains, and these differ depending on the Ethiopian region targeted for introduction. Decisions regarding which strain to keep in which respective environment can be aided by outcomes of statistical models used in this study. However, the final advice regarding which strain is preferable in which Ethiopian region needs to consider local farmer preferences and other performance traits.

**Abstract:**

Ethiopia is a developing nation that could highly benefit from securing food via improved smallholder poultry farming. To support farmer and breeding decisions regarding which chicken strain to use in which Ethiopian environment, G*E analyses for body weight (BW) of growing male and female chickens were conducted. Research questions were (1) if a G*E is present for BW and (2) which strain performs best in which environment in terms of predicted BW. Analyses were performed using predicted BW at four different ages (90, 120, 150, and 180 days) of five strains (Horro, Koekoek, Kuroiler, Sasso-Rhode Island Red (S-RIR), and Sasso) tested in five Ethiopian regions (Addis Ababa, Amhara, Oromia, South Region, and Tigray) that are part of three Agro-Ecological Zones (AEZ) (cool humid, cool sub-humid, and warm semi-arid). The indigenous Horro strain was used as a control group to compare four other introduced tropically adapted strains. The dataset consisted of 999 female and 989 male farm-average BW measurements. G*E was strongly present (*p* < 0.001) for all combinations of strain and region analyzed. In line with previous research, Sasso was shown to have the highest predicted BW, especially at an early age, followed by Kuroiler. Horro had the lowest predicted BW at most ages and in most regions, potentially due to its young breeding program. The highest predicted BW were observed in Tigray, Oromia, and Amhara regions, which are in the main part of the cool sub-humid AEZ.

## 1. Introduction

Poultry farming is a significant farming activity in African countries and contributes positively to the socio-economic livelihoods of families in these nations [1,2]. Livestock contributes 38.5% to the income of Ethiopian poultry-keeping households [3]. The global demand for animal protein is rising and is expected to increase by 70–80% between 2012 and 2050, with poultry expected to have a bigger increase in production than other livestock [4,5,6]. Multidisciplinary research supporting the local adaptation and tailoring of sustainable poultry production for Ethiopian smallholders is thus needed [7]. Hence, looking for local African smallholder poultry production system optimization strategies is important. The African Chicken Genetic Gains (ACGG) project (https://africacgg.net/, accessed on 27 August 2021), led by the International Livestock Research Institute located in Addis, Ethiopia, aims to achieve this optimization [8,9]. One of the strategies was the introduction and testing of various tropically adapted dual-purpose poultry strains, which are more productive than local strains, into different agro-ecologies [8,9]. To determine the benefits of this approach, it is important to know whether these new environments have an effect on the production traits of the introduced strains.

Strain suitability was successfully identified for different environments with ACGC data by applying an ecological niche modelling approach based on Geographic Information Systems, which were only recently applied in livestock research [10]. Body weights (BW) were predicted (during the growing period: day 100–135 and adult period: day 140–505) using climate data from the different agro-ecologies in Ethiopia and local performance data from five dual-purpose strains (Horro, Koekoek, Kuroiler, Sasso-Rhode Island Red (S-RIR), and Sasso) [11]. Horro is an indigenous Ethiopian strain improved by a breeding program implemented in 2008 [12]. The other four strains were introduced to Ethiopia. Based on these data from five Ethiopian regions (Addis Ababa, Amhara, Oromia, Southern Nations Nationalities and People’s Region (SNNP) in short Southern Region, and Tigray), the mean estimated male BW in the growing phase were highest for Sasso, followed by Kuroiler [11]. The mean estimated female BW were highest for Koekoek, followed by Sasso. The highest BW for these strains were found in Tigray for male and female Sasso, in Oromia for male Kuroiler, and in Amhara for female Koekoek. These general results are the basis of a hypothesis for the current study. However, those results were based on phenotypic distribution models [11], while the current study uses classical genotype by environment interaction (G*E) analysis.

In classical G*E analyses, the relative change in phenotypic performance of two or more genotypes is measured in different environments [13,14]. G*E can be useful for comparing African chickens in environments such as housing systems or seasons [15]. A G*E effect can occur when the environments of breeding (where animals are selected) versus production differ, and its magnitude can be so large that accounting for it is needed by different breeding programs [13,16]. This problem is highly prominent in poultry breeding in developing countries with various studies on African chickens kept under intensive research stations versus extensive smallholder farms, showing the clear presence of G*E [17,18,19]. The current study tries to avoid this bias in performance results by using data collected on smallholder farms only. Since the current study was on purebred and crossbred chickens, genotypes are referred to as strains, and G*E is referred to as strain by environment interaction (S*E). Horro, as the best performing local strain, can be considered a control group for performance comparison to the other four introduced strains. While predicting S*E, environments can be defined as the Ethiopian regions tested in the ACGG project or different agro-ecologies, with relative strain performance compared between environments. Ranking outcomes can be direct advice for deciding which strain(s) to use for poultry body weight performance in the five Ethiopian regions.

The largest advantage of classic G*E with ACGG data is the possibility for clear result-based decision making, i.e., a farmer in an Ethiopian region knows which of the tested strains most likely performs best in this region. This is in contrast to the phenotypic distribution modelling results of Lozano-Jaramillo et al. (2019) [11], which show performance prediction differentiation in certain areas on a heatmap of the country. These are scientifically interesting results, but significantly less practical to use for coming up with solid advice for breeding strategies compared to what can be achieved with our classic G*E results. The aim of the current study was to perform an S*E analysis using BW during the growth period as the production trait. Therefore, the goal was to obtain novel predicted performance results, which are highly relevant due to direct applicability, while making decisions regarding ACGG strain preferences in various Ethiopian environments. The two key research questions were as follows

(1)Is S*E present, i.e., do different strains react differently to the various Ethiopian environments;(2)Which strain performs best in which environment in terms of predicted BW.

As BW performance information allows farmers to make choices between strains for their specific environment, the latter question is of substantial importance.

## 2. Materials and Methods

### 2.1. Study Design and Data Collection

Five chicken strains (Horro, Koekoek, Kuroiler, S-RIR and Sasso) were placed and tested in 63 villages in five Ethiopian regions (Addis Ababa (AA), Amhara (AM), Oromia (OM), South Region (SR), and Tigray (TG)). The local Horro strain was used as the control for comparison with the four introduced tropically adapted strains. Six-week (~50 day) old chicks were placed in 1393 households with approximately 25 chicks of one strain per household, while each of the different strains was present in each village. Therefore, the study was based on an initial number of approximately 35 birds distributed. However, the realistic approximate number of birds that the results of this study are based on is considerably lower, due to various reasons such as farming practices, data measurement techniques, and data cleaning. To test the performance of the introduced strains under the same low-input conditions as indigenous chickens, they were placed among these originally present chickens in farmer households. Only one introduced strain was tested in each household. Similarly, as for the indigenous chickens, feed intake was mainly from scavenging, with supplemented feed given at the farmer’s discretion. Supplemented feed was either grown by the farmers or commercially sourced. In an attempt to standardize farm conditions, training was provided to all farmers regarding supplementary feeding of the introduced birds, as well as for building a night shelter. Distribution started in August 2016 and data collection ended in January 2018 [11]. Data on total live BW and the number of birds were collected as a group measurement. Every two weeks, group weights and counts were taken separately for male and female birds. Weight data of male chickens were collected until 20 weeks (~140 days) or until they reached two kilograms because males are traditionally sold when they reach this BW, which is around this age, while female chickens are kept longer for egg production. Hence, weight data collection took place until 72 weeks (~505 days) for females, which caused the data collection period to be approximately 18 months in total [11]. Regarding ethical considerations, the study did not include in vivo animal experiments or laboratory animals. Hence, the current study complies with the ARRIVE guidelines [20].

### 2.2. Strain Descriptions

#### 2.2.1. Horro

Horro is the only native Ethiopian strain out of the five, belonging to an indigenous population in the Horro district located in the Oromia region, in the cool wet western highlands of the country [11,21]. The local chicken strain is the dominant type in the Horro district (only 2.8 to 7.2% being crossbred) kept mainly under scavenging and low-input conditions [22]. The African Chicken Genetic Gains (ACGG) project (https://africacgg.net/, accessed on 27 August 2021) is the first attempt to look into the adaptation possibilities of the strain to an on-farm environment and management conditions [11]. The breeding program was established in 2008 only and mainly focuses on the genetical improvement in growth traits, as this is preferred by local smallholder farmers [12,21]. It is found that BW traits (around 115 days of age) have a strong genetic correlation with egg production (0.92, 0.69, and 0.73 for the cumulative number of eggs produced from months 1 to 2, 3 to 6, and 1 to 6, respectively), making Horro a very promising dual-purpose strain to improve via breeding [21].

#### 2.2.2. Koekoek

Koekoek is a South African strain originating from a crossbred of three different breeds [11]. This crossbred was established in the 1950s at the Potchefstroom Agricultural College using Black Australorp, White Leghorn, and Barred Plymouth Rock as breeds in the cross [23]. Therefore, the strain is also called Potchefstroom Koekoek in its complete name and can be considered a locally developed strain. Roosters and culled hens are usually used for meat production. But the strain is also very popular amongst rural farmers from South Africa and neighbouring countries for meat as well as egg production, their broodiness, and hatching their own offspring [11,23]. Based on GIS analysis, 13% of Amhara would be the most suitable Ethiopian region for Koekoek, followed by 11% Oromia, 10% South Region, and just 0.75% Tigray [24]. This is due to Koekoek surviving better in areas with colder temperatures with larger annual temperature fluctuations [24].

#### 2.2.3. Kuroiler

This is a dual-purpose strain developed under humid conditions by Keggfarms in India to perform in low-maintenance systems [11]. This commercial hybrid chicken was made by crossing Rhode Island Red (RIR) females with either coloured broiler males or White Leghorn males [23]. With GIS analysis, it was shown that the BW distribution of Kuroiler was mostly influenced by environmental variables including precipitation [11]. This supports the humid origin of Kuroiler and the statement that Ethiopian farmers prefer precipitation such as rain causing higher vegetation and therefore lower predation of their chickens [7,11].

#### 2.2.4. S-RIR and Sasso

Sasso is a commercial strain originating from warm and dry areas in Southern France where it was developed by the breeding company SASSO [11,23]. According to GIS analysis, Sasso weight performance is linked to temperature-associated variables supporting its warm environmental origin [11]. S-RIR is a crossbred of Sasso and RIR specifically made for the ACGG project on a private farm [25]. Therefore, there are rarely any other studies present about S-RIR performance evaluation under scavenging conditions [11]. Diverse environmental variables were associated with BW distribution of S-RIR at different ages, suggesting the strain’s response to the environment depends on age [11]. Purebred Sasso is predicted to be heavier than S-RIR during a growing period of 100–135 days [11]. Both strains outperformed the other ACGG strains (Horro, Koekoek, and Kuroiler), indicating they can deal with low-input conditions and are interesting for further on-farm testing in Ethiopia [11].

### 2.3. Area Descriptions

Across the five Ethiopian regions, three broadly different Agro-Ecological Zones (AEZ) were present (cool humid, cool sub-humid, and warm semi-arid). General details on the climates of the five Ethiopian regions and their AEZ are described by using average annual values in Table 1, which is useful for an overall understanding of the backgrounds of each region. Data on various weather parameters at the time of the experiment are displayed in Figure 1 and Table 2, which are both used to interpret the results of this study. Data from this figure and table were requested from the National Meteorology Agency (NMA) of Ethiopia (http://ethiomet.gov.et/, accessed on 2 February 2022). NMA provided data regarding temperature and precipitation, obtained from at least one or two weather stations located in, or very near to, the districts that the chicken strains were placed during the experiment. These districts are part of the Ethiopian regions in the study, and more information regarding them can be found on the website (https://africacgg.net/ethiopia/, accessed on 27 August 2021). Averages of data from all weather stations in a particular Ethiopian region were calculated and plotted per month of the experiment. Averages of weather parameters per Ethiopian region over the whole duration of the experiment (18 months) are provided in Table 2, just as the elevation of all weather stations per Ethiopian region and number of weather stations the data are based on.

### 2.4. Data Cleaning

All measurements with ages below 50 days and above 300 days were excluded before calculating the farm-average BW for males and females at each measurement time and each household by dividing the total live BW by the number of birds per weighed group. Average BW values between 50 and 6000 g were kept for further analysis. BW at four different ages (90, 120, 150, and 180 days) were predicted via linear interpolation between adjacent weighing ages. The cleaned dataset contained a total of 1988 farm-average BW measurements, of which 999 were females and 989 were males. Data were corrected for a decreased number of birds per household due to not being able to collect all birds free ranging in the field, but also mortality, birds being sold, or other potential reasons. The number of birds per household was recorded at each group weighting. In the cleaned dataset, the original number of 25 birds per household decreased to an average of 8.4 females and 7.2 males (Table 3). This means that with female and male birds in 999 and 989 households, respectively, the study was based on approximately 8392 female and 7097 male birds. Table 3 displays more information regarding the structure of the cleaned dataset, including the number of villages, households, and birds per household. The number of households, per five strains, per age, per environment (five Ethiopian regions or three AEZ), and per sex are the numbers of records eventually used for analysis and are given in Appendix A. Predicted BW at 90, 120, 150, and 180 days of age were checked for normality by making Q-Q plots of these BW within the regions and AEZ. These plots were used to remove outliers and retain observations within the range of growth weights mentioned in literature for the local African or introduced strains at each specific age with the thresholds provided in Table 4 [28,29]. Ranges of final numbers of households with BW at the different ages that were used for analyses are displayed per strain, per sex, and per environment in Table 5. There is little variation in the values, giving support to the robustness of the study.

### 2.5. Statistical Analysis

A linear fixed effects model was implemented using PROC GLM, SAS version 9.4 [30]. For complementary evidence, two models were used to test the effect of the S*E on the predicted BW phenotypes at four different ages. The models were (1) a simple model with the effects of strain and environment and their interaction and (2) a complex model that also included the sex of the birds and all types of interaction.
y_ijk_ = μ + Strain_i_ + Environment_j_ + Sn*E_ij_ + e_ijk_(1)
y_ijkl_ = μ + Strain_i_ + Environment_j_ + Sex_k_ + Sn*E_ij_ + Sn*Sx_ik_ + E*Sx_jk_ + Sn*E*Sx_ijk_ + e_ijkl_(2)

y_ijk(l)_ is the predicted BW at either 90, 120, 150, or 180 days of age; *µ* is the mean; Strain_i_ is the fixed effect of genotype or strain (*n* = 5); Environment_j_ is the fixed effect of the environment, either the Ethiopian region (*n* = 5) or AEZ (*n* = 3); Sex_k_ is the fixed effect of sex (*n* = 2); Sn*E_ij_ is the fixed effect of strain by environment pairwise interaction, so the S*E; Sn*Sx_ik_ is the strain by sex interaction; E*Sx_jk_ is the environment by sex interaction; Sn*E*Sx_ijk_ is the triple interaction between all three main fixed effects; and e_ijk(l)_ is the random residual assumed to be ~N(0,I σ^2^_e_), with I being an identity matrix and σ^2^_e_ being the residual variance.

Significance tests of the main as well as interaction components in models (1) and (2) were carried out using ANOVA. Model (2) is the most complete version of the complex model, prior to the exclusion of non-significant effects from the model. This complete model was adjusted in a backward stepwise manner to include only significant effects. The non-significant interaction effect with the highest *p*-value was excluded first, and the model was run again. If non-significant effects were found, the effect with the highest *p*-value was again excluded. Non-significant main effects were kept if they were part of a significant interaction effect. In case of a significant three-way interaction, the full model was kept. Final complex models are displayed per age in the results section. Graphs showing the S*E were produced by plotting the strain’s predicted BW, using the model derived from (2), against the environments (Ethiopian region or AEZ) for each sex at each age separately.

The Sasso strain was not tested in AA and no females were present in TG. BW of females was predicted in TG based on male data if no region-by-sex effect was present. Note that the pairwise comparisons of least squares mean for strains and environments (both regions and AEZ) could not be performed because of the missingness of subgroups (e.g., no Sasso in AA). Statements of differences of such subgroups will therefore be approximate. Appendix A provides means and standard deviations of predicted BW per sub-group (strain, age, sex, and environment), which is an interpretation of strain performance rankings between environments for each age and sex category. This is additional strain ranking information for the S*E graphs in the results section.

## 3. Results

### 3.1. Simple S*E Model

Based on the results of model (1), the effects of strain, environment (alternatively region or AEZ), and their interaction were all strong (*p* < 0.001) for all age classes (90, 120, 150, and 180 days). This was the case when taking either Ethiopian regions or AEZ as the environment. Therefore, S*E was present in all cases.

### 3.2. Complex S*E Model for Ethiopian Regions

The complex model (2) with strain, environment, sex, and all their two-way and three-way interactions showed highly significant S*E effects (*p* < 0.001; bold in Table 6) when using the Ethiopian region as the environment. The effects of strain, environment, and sex were significant at all ages. *p*-values of other effects that were significant in the final model for predicted BW at different ages are shown in Table 6, exposing the eventual structure of those models.

With the Ethiopian region as the environment, the S*E plots show that Sasso has the highest predicted BW in most environments and at most ages, especially in the OM, AM, and TG regions (Figure 2). In the AA region, Sasso was not tested, and in the SR region, Sasso shows an average BW compared to other strains. In the AA region, high predicted BW is seen for S-RIR (at 90 and 180 days) and Kuroiler (at 120 and 150 days). The control strain Horro had the lowest predicted BW in most regions at all ages. In SR, the differences of predicted BW between strains were smaller compared to other regions. TG had the highest predicted BW at most ages.

### 3.3. Complex S*E Model for AEZ

S*E effects (*p* < 0.001; bold in Table 7) were found at all ages with the complex model (2) when using AEZ as the environment. The effects that were kept in these models are shown in Table 7 with their respective *p*-values.

At an early age (90 days), Sasso has the highest predicted BW across the AEZ with the exception of male Kuroiler in cool humid AEZ (Figure 3). At later ages (120, 150, and 180 days), the female Sasso remained the heaviest in the cool humid and cool sub-humid AEZ. Also, Kuroiler, Koekoek, and S-RIR perform similarly to Sasso, especially for females in cool humid AEZ. Particularly at age 90 days, the performance of strains was more similar in cool sub-humid compared to cool humid. The average predicted BW was the highest in cool sub-humid. At later ages (120 and 150 days), the performance for females was more variable in cool sub-humid AEZ and the average predicted BW at age 180 days in cool sub-humid was lower than for cool humid. Horro, as a control group, had the lowest predicted BW in most AEZ, apart from Koekoek at some ages and AEZ. Kuroiler, S-RIR, and Sasso were not tested in warm semi-arid AEZ. In this AEZ, the BW of Koekoek and Horro are similar and always lower than in cool sub-humid. In contrast to warm semi-arid, the largest performance difference was between Koekoek and Horro in cool humid.

## 4. Discussion

S*E was shown to be present in simple and complex models (Table 6 and Table 7; *p* < 0.001), predicting BW of five different strains across five different regions of Ethiopia that represent three AEZ. Hence, we can say with certainty that by using multiple models, it is confirmed that different strains do react differently to the environments of the various Ethiopian regions or AEZ at all tested ages. Which strain performs best in which environment in terms of predicted BW can be answered by consulting the S*E plots based on final complex models. This is the most important question for Ethiopian smallholder farmers located in specific environments.

Primarily, Sasso had the highest predicted BW in OM, AM, and TG regions, and Kuroiler had the highest predicted BW in AA (Figure 2). As explained in the introduction, the current results using the Ethiopian region as an environment can be validated by the hypothesis results in the GIS analysis study since they used almost the same dataset but a very different analysis approach [11]. Here, predicted male BW in the growing phase (day 100–135) was highest for Sasso, followed by Kuroiler, and Sasso performed best in TG, which is the region with the highest predicted BW. It is worthy of note that current results on TG could be less reliable than on other regions since female predictions are based on the male-only measurements in the absence of region-by-sex effects while using minimal data (Table 5; *n* = 175–177). Hypothesized results on predicted female BW differed slightly from current results, as it was Koekoek followed by Sasso with the highest [11]. Further validation of our study are findings that female BW was highest for Sasso, followed by Kuroiler, while measured on-farm in Ethiopia [31]. Additionally, both studies predicted Horro to have the lowest BW in all cases [11,31]. Using AEZ as the environment in S*E analysis, the control Horro and Sasso had generally the lowest and highest predicted BW, respectively, in all environments.

Horro not performing well can be explained by the limited breeding program, having only been established in 2008, with limited time to genetically improve this indigenous strain [12,21]. BW and other traits did improve in Horro after six or seven generations of breeding, making the program successful [12,32]. However, improved Horro still performs lower than the other four ACGG strains if tested on-farm in two districts in OM [32]. SR was the only region with a predicted BW of Horro not being the lowest. In SR, the differences in predicted BW between all strains were minimal (Figure 2). Clear explanations for this performance similarity are lacking. SR being a climatically diverse region included in two AEZ being cool humid and cool sub-humid could have introduced variation in performance (Table 1). Sufficient data in SR (Table 5; *n* = 403–409) could maybe explain the recorded performance similarity, which potentially could have been observed if more data were available as it is for regions such as OM and TG (Table 5; *n* = 241–252, *n* = 175–177, respectively).

Sasso or Kuroiler having the highest predicted BW in many of the environments is not surprising, as both strains have been classified as better suited for single-purpose meat production compared to dual-purpose meat and eggs usage while tested on-station with four other strains in Nigeria [33]. On-farm testing in Uganda showed that Kuroiler significantly outperformed indigenous chicken strains, indicating the strain can easily adapt to scavenging conditions [34]. Kuroiler was also the heaviest, followed by S-RIR, Koekoek, and lastly, the lightest Horro which were all kept at two experimental stations in Ethiopia [35]. An additional hypothesis predicted Sasso and S-RIR to outperform the other ACGG strains, possibly because they can deal with low-input conditions [11]. Therefore, the Sasso and Kuroiler strains, and possibly the S-RIR strains could be subjects of further on-farm testing in Ethiopia [11].

Sasso and Kuroiler perform well in TG, OM, and AM (Figure 2). TG is the warmest, driest (Figure 1 and Table 2), and most northern region, completely in a cool sub-humid AEZ, with the highest average predicted BW of all strains. A potential benefit of a drier environment is lower disease risk [7]. However, the lower reliability of predictions in TG due to data from males only should not be dismissed. OM is the largest region located centrally in Ethiopia. OM surrounds the capital, which is the cool humid AA region. OM has great environmental variability with cool sub-humid AEZ and warm semi-arid AEZ (Table 1) [26]. Hence, the average weather data per month (Figure 1) can be less reliable since it is an average for this very extensive and diverse region. Predictions in warm semi-arid AEZ could be less reliable due to minimal data (Table 5; *n* = 89–92). Other studies confirm climatic variability of OM, while comparing poultry keeping among districts within OM [7,22,36]. It is shown that village poultry performance in the high-altitude Horro district where the indigenous Horro strain originates from differs from other districts such as Ada and Jarso [7,22]. Hence, Sasso and Kuroiler strains could potentially perform differently among districts within OM. AM, located in between TG and OM, has much more climatic diversity, with parts of it belonging to the cool humid AEZ and others to the cool sub-humid AEZ. AM is very humid and receives the highest percentage (80%) of total rainfall in Ethiopia [26] with a high annual rainfall of 1110.81 mm per year (Table 1) [27]. This pattern of highest precipitation in AM is also clearly visible during the experiment in Figure 1B.

Koekoek and Kuroiler performing well in AM, especially at a later age (Figure 2), is supported by studies predicting Koekoek would have highest BW in AM [11] and 13% of AM predicted to be best suitable for Koekoek [24]. This was explained by major annual temperature fluctuations in AM and claiming that high precipitation is preferred by farmers for chicken rearing, due to more vegetation and reduced chicken predation [7]. Again, it is precipitation variables predicted to influence BW distributions of Kuroiler that can be explained by its humid Indian origin [11]. AM does not appear to have different or more extensive temperature ranges in comparison to other regions (Figure 1A and Table 1), indicating that high precipitation in AM may be the most likely climatic explanation for high Koekoek and Kuroiler performance. AM, just like OM and TG, are all part of the cool sub-humid AEZ (Table 1), typically having the highest performances (Figure 3), which agrees with the high performance in the regions that include this AEZ. More climatic diversity can be present within a single Ethiopian region while the climate is more similar within one type of AEZ. Hence, analysis per AEZ is preferred for scientific interpretations. However, the results of the study design by geographical region can be shaped towards clear advice for local farmers about which strain is preferred per region which adds useful information for making breeding decisions. Just as annual weather and climate values in Table 1 are useful for decision making regarding where to place which strain in the future, weather values during the experiment (Figure 1 and Table 2) are useful for finding correlations and explanations of the predicted BW data.

Apart from deciding between strains solely based on BW performance data, other factors need to be considered, such as the social context of strain preference by local smallholder farmers, other relevant traits, and uncertainties related to the difficulty of collecting on-farm data. Different traits and breeding practices are preferred per the Ethiopian region, but the ultimate breeding goal would be to develop a dual-purpose strain based on indigenous chicken resources [37]. Introducing improved Horro would be most preferred by the farmer target group whose main income comes from livestock [37,38]. However, it is stated that the additional cost of interventions needed to achieve higher productivity of improved Horro is not offset by higher income for poultry farmers in OM [38]. This is contradictory to a production and marketing survey in Ethiopia, Nigeria, and Tanzania suggesting that integrating improved dual-purpose strains in village production systems would improve the income of producers, enhance eggs and live chicken supply, and generate employment opportunities for the rural youth and other marketing actors [39]. Currently, the proportion of indigenous chickens has decreased and the proportion of exotic chickens has increased among Ethiopian poultry smallholder farmers [40].

In addition to BW or growth traits, egg sales and high egg-producing traits in chickens are very important to African chicken smallholder farmers [3,40]. Many other traits are important to estimate strain performance in on-farm research in Ethiopia, such as egg number, egg weight or other laying traits, survival rates, feed intake (as efficiency indication), fertility, and hatchability traits [12,31,32,36,41,42]. Smallholder on-farm data collection for breeding can be problematic, as conditions such as large populations, recording of performance and pedigree, and recording environmental variation are lacking [10,37,43]. The risk of smallholder farmers dropping out of research is relatively high, due to high chicken mortality or lack of motivation [32]. Potential limitations in our study are only one strain type being present per household and the realistic approximate number of birds being lower than the initial 25 distributed chicks (Table 3). Lacking conditions may be solved by on-station breeding data collection; however, this often suffers from G*E with on-farm chickens having significantly lower performance (mainly weight gain traits) compared to on-station [17,18,19]. This appears to happen when comparing current on-farm results to available results of four ACGG strains (Horro, Koekoek, Kuroiler, and SRIR) tested at two research stations located in Ethiopia [35]. The mean predicted BW at day 120 (Appendix A) were all lower than the mean predicted BW at week 17 (~120 days) [35] (Table 4 in the on-station data publication). The only exception was a higher mean predicted BW of Kuroiler in AA, but this could be unreliable due to limited data (Appendix A; *n* = 7 for females and *n* = 1 for males). Therefore, it seems that the overall data collection methods have a high influence since the full potential on-station performance is higher than the low-input on-farm performance assessed in our study. It is recommended that Ethiopia should invest in proper data collection to achieve faster genetic gains [44].

## 5. Conclusions

Regarding research question (1) about S*E presence, the ACGG strains do react differently to the different environments of Ethiopian regions or AEZ in terms of predicted BW. The presence of S*E was shown substantial (*p* < 0.001) with both a simple and more complex model at various ages (90, 120, 150 and 180 days). Regarding research question (2) about the strain performing best in the Ethiopian region, it can be concluded that Sasso performed best in most situations, especially in OM, AM, and TG, followed by Kuroiler. The improved Horro control group commonly performed lowest in most of the test environments, probably due to the indigenous strain being the product of a very young breeding program. SR was a clear exception with Horro not having the highest BW. The current findings are in line with previous research. TG is the region with approximately the highest predicted BW. Predicted BW is also high in OM and AM. TG, OM, and AM are all part of the cool sub-humid AEZ, which was the AEZ having mostly highest BW performances. High precipitation in the AM could explain Koekoek and Kuroiler performing well in the AM at a later age. The study design was per Ethiopian region to ease decision making for farmers and breeding. However, analysis per AEZ is scientifically preferred due to climatic similarity. Regarding research question (2) about the strain performing best in AEZ, the conclusion can be that at early ages Sasso has high predicted BW in most AEZ. At later ages, female Sasso and male Kuroiler perform well in cool sub-humid AEZ, while male Koekoek and S-RIR also perform well in cool humid AEZ.

The findings of this study are a good indication of breed preferences based on BW in certain Ethiopian regions or agro-ecologies. Outcomes could be used by local farmers who keep chickens, as well as breeders who want to further improve certain breeds for various conditions. The on-farm data collection of this study generates a unique, highly valuable, but also limited dataset. Future research could try to overcome these limitations by either upscaling the sample size, which will be highly labour intensive, or by conducting research station data collection. However, the latter is expected to result in G*E presence, in which chickens have higher performances on-station compared to if the chickens were kept on-farm, while this on-farm environment is where chickens eventually must be productive in.

## Figures and Tables

**Figure 1 animals-13-03121-f001:**
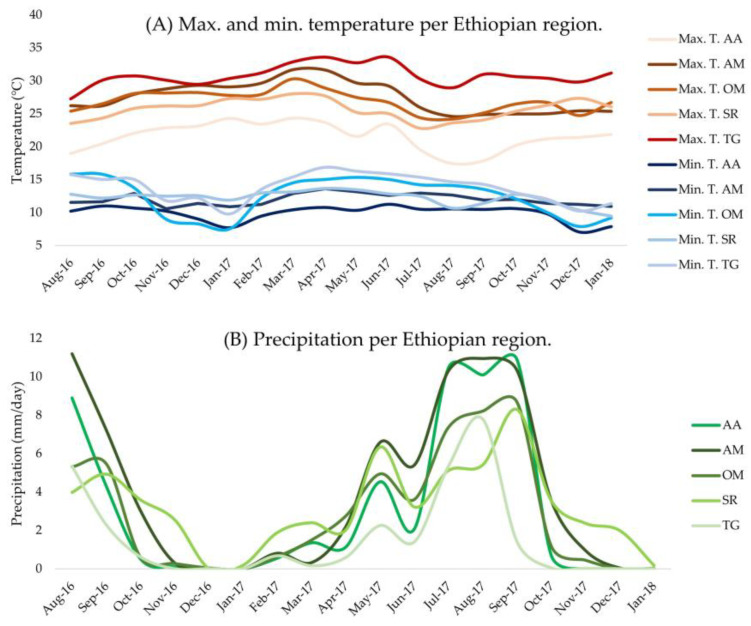
Weather data per Ethiopian region (AA, AM, OM, SR, and TG) at the time of the experiment (August 2016 to January 2018). (**A**) Average maximum and minimum temperature in °C per month, measured each day at 18:00 and 9:00 for max. and min. temperature, respectively. (**B**) Precipitation in millimetre (mm) measured each day at 9:00, converted to a monthly average of mm per day. Averages of the plotted data and information regarding the weather stations that the data are obtained from are given in Table 2. All data are obtained as requested from the National Meteorology Agency (NMA) (http://ethiomet.gov.et/, accessed on 2 February 2022).

**Figure 2 animals-13-03121-f002:**
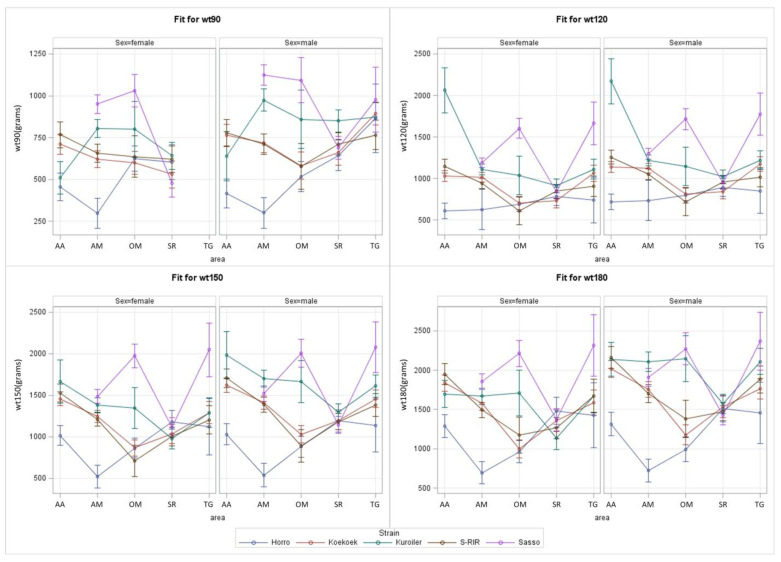
S*E plots made using derived models from the complex model (2) (effects given in Table 6) displayed per sex (female or male) and per predicted BW in grams at a certain age (wt90 is BW at 90 days, wt120 is BW at 120 days, wt150 is BW at 150 days, and wt180 is BW at 180 days). Environments were Ethiopian region (AA, AM, OM, SR, and TG). The 95% confidence limits are shown in plots.

**Figure 3 animals-13-03121-f003:**
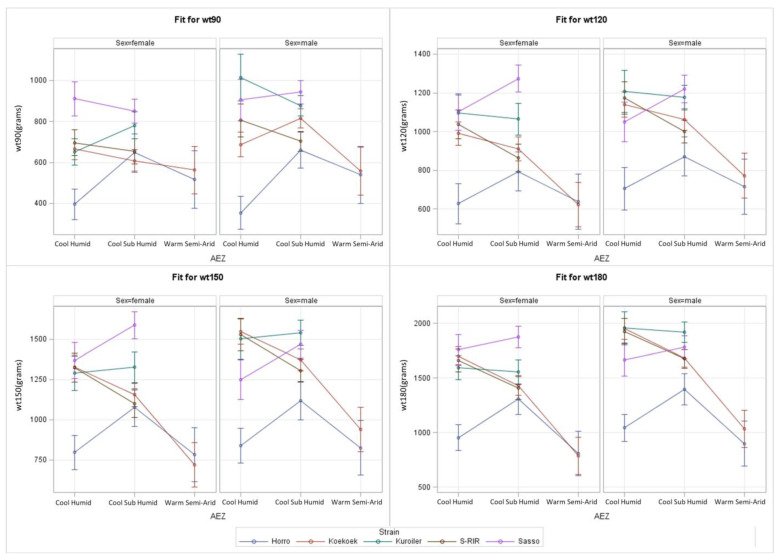
S*E plots made using derived models from complex model (2) (effects given in Table 7) displayed per sex (female or male) and per predicted BW in grams at a certain age (wt90 is BW at 90 days, wt120 is BW at 120 days, wt150 is BW at 150 days, and wt180 is BW at 180 days). Environments were AEZ. The 95% confidence limits are shown in plots.

**Table 1 animals-13-03121-t001:** General climate description of five Ethiopian regions (Addis Ababa (AA), Amhara (AM), Oromia (OM), South Region (SR), and Tigray (TG)), among which three Agro-Ecological Zones (AEZ) are present (cool humid, cool sub-humid, and warm semi-arid) based on the government of Ethiopia [26]. Mean annual temperature, rainfall, altitudes, and AEZ are given. Annual rainfall in mm of AM is obtained from [27]. Large variations are due to the presence of climatic and geographic diversity within Ethiopian regions.

Region	AA	AM	OM	SR	TG
Climate description	Mild, Afro-Alpine temp-, warm temperate	25% cool to cold, 44% warm to cool, 31% warm to hot	Big diversity; dry, tropical rainy climate, temperate rainy	56% hottest lowland, 44% temperate	39% semi-arid, 49% warm temp-, 12% temperate
Temp. (°C)	9.9–24.6	15–21	18–39	15–30	-
Rain (mm)	-	80% Ethiopia 1111	410–2000	500–2200	450–980
Altitude (m)	2200–2500	High > 1500, Low 500–1500	>500–4607	376–4207 56% < 1500	600–2700
AEZ	Cool humid	Cool humid, cool sub-humid	Cool sub-humid, warm semi-arid	Cool humid, cool sub-humid	Cool sub-humid

**Table 2 animals-13-03121-t002:** Averages for parameters with weather data over the total duration of the experiment, i.e., average over the 18 months of August 2016 to January 2018. Data are given per Ethiopian region (AA, AM, OM, SR, and TG) with the number of weather stations the data are based on provided per parameter and per Ethiopian region at the bottom of the table. The parametric average elevation of weather stations given in meters (m), average temperature in °C measured daily at 18:00 for maximum and at 9:00 for minimum temperature, average precipitation in millimetre per day (mm/day) measured daily at 9:00 and average % relative humidity with the average per day calculated based on five time points (6:00, 9:00, 12:00, 15:00, and 18:00). All data are obtained from the National Meteorology Agency (NMA) (http://ethiomet.gov.et/, accessed on 2 February 2022).

Ethiopian Region	Elevation (m)	Max. T. (°C)	Min. T. (°C)	Precipitation (mm/day)	Relative Humidity (%)
AA	2548	21.5	9.9	3.1	56
AM	1936	27.5	12.0	4.1	-
OM	1707	26.9	12.4	2.8	-
SR	2032	25.7	12.2	3.2	67
TG	1717	30.8	13.8	1.5	-
	Number of weather stations data are obtained from				
AA	3	3	3	3	2
AM	9	5	5	10	-
OM	7	5	5	7	-
SR	7	5	5	7	1
TG	4	4	4	5	-

**Table 3 animals-13-03121-t003:** Values left in the dataset after cleaning. Number of villages per Ethiopian region (ER) (AA, AM, OM, SR, and TG) and per AEZ (cool humid, cool sub-humid, and warm semi-arid) they are located in. N is the number of households that were given a certain strain (Horro, Koekoek, Kuroiler, Sasso-Rhode Island Red (S-RIR), and Sasso). Average number of birds per household (hh) based on all group weight attempts. Means of this value are given per strain. All data are displayed per sex (female and male).

Female			Male		
Villages	ER	AEZ	Villages	ER	AEZ
6	AA	cool humid	6	AA	cool humid
7	AM	cool sub-humid	7	AM	cool sub-humid
7		cool humid	7		cool humid
10	OM	cool sub-humid	6	OM	cool sub-humid
3		warm semi-arid	3		warm semi-arid
5	SR	cool sub-humid	1	SR	cool humid
			11		cool sub-humid
			8	TG	cool sub-humid
38	Tot		49	Tot	
	N	Birds per hh		N	Birds per hh
Horro	157	8.1	Horro	139	6.8
Koekoek	288	9.8	Koekoek	303	8.2
Kuroiler	185	5.9	Kuroiler	178	5.5
S-RIR	196	10.4	S-RIR	222	6.9
Sasso	173	6.7	Sasso	147	7.8
Tot	999	8.4	Tot	989	7.2

**Table 4 animals-13-03121-t004:** Thresholds used to clean data of predicted BW at four different ages. Minimum and Maximum values are given in grams.

Age BW (Days)	Min (g)	Max (g)
90	200	2500
120	300	3000
150	400	3500
180	500	4000

**Table 5 animals-13-03121-t005:** Ranges of numbers of households analyzed per strain (Horro, Koekoek, Kuroiler, S-RIR, and Sasso), per sex (F is female and M is male) and per environment (Ethiopian regions or AEZ). Five Ethiopian regions (AA, AM, OM, SR, and TG) and three AEZ (cool humid, cool sub-humid, and warm semi-arid) are analyzed. Ranges are given for the number of households with data available at four different ages (90, 120, 150, and 180 days).

Region	AA				AM				OM				SR		TG	
Sex	F		M		F		M		F		M		F	M	F	M
Horro	37–39		32		7–41		3–43		44–47		26–29		17–23	28–30	0	8
Koekoek	82		61–63		105–108		81–85		61–62		35–37		33–35	36–39	0	73
Kuroiler	7–40		0–2		96–100		59–62		7–8		4–7		31–35	59–63	0	44
S-RIR	43		39–40		95–103		63–65		13–16		8		32–33	63–65	0	41–43
Sasso	0		0		95–98		71–72		31–40		4–5		32–35	58–61	0	9
AEZ	cool humid						cool sub-humid						warm semi-arid			
Sex	F			M			F			M			F		M	
Horro	44–79			35–66			45–48			44–48			19		18–19	
Koekoek	135–138			100–102			112–122			164–167			26–27		26–27	
Kuroiler	60–96			28–30			81–86			142–146			0		0	
S-RIR	87–94			57–58			96–101			159–163			0		0	
Sasso	52			33–36			106–121			109–111			0		0	

**Table 6 animals-13-03121-t006:** *p*-values of effects (R*Sn is Region by Strain, R*Sx is Region by Sex, Sn*Sx is Strain by Sex, and R*Sn*Sx is Region by Strain by Sex) kept in the final complex model (2) for ages (90, 120, 150, or 180 days) using Ethiopian region as the environment. *p*-values of the strain by environment interaction (S*E) effect, in this case Region by Strain effect, are displayed in bold.

Effect:	Region	Strain	Sex	R*Sn	R*Sx	Sn*Sx	R*Sn*Sx
90 days	<0.001	<0.001	<0.001	**<0.001**	0.042	0.007	-
120 days	<0.001	<0.001	<0.001	**<0.001**	-	-	-
150 days	<0.001	<0.001	<0.001	**<0.001**	-	0.001	-
180 days	<0.001	<0.001	<0.001	**<0.001**	-	<0.001	-

**Table 7 animals-13-03121-t007:** *p*-values of effects (AEZ, A*Sn is AEZ by Strain, A*Sx is AEZ by Sex, Sn*Sx is Strain by Sex, and A*Sn*Sx is AEZ by Strain by Sex) kept in the final complex model (2) for ages (90, 120, 150, or 180 days) using AEZ as environment. *p*-values of the S*E effect, in this case, AEZ by Strain effect, are displayed in bold.

Effect	AEZ	Strain	Sex	A*Sn	A*Sx	Sn*Sx	A*Sn*Sx
90 days	0.006	<0.001	<0.001	**<0.001**	0.842	0.002	<0.001
120 days	<0.001	<0.001	<0.001	**<0.001**	-	0.009	-
150 days	<0.001	<0.001	<0.001	**<0.001**	-	<0.001	-
180 days	<0.001	<0.001	<0.001	**<0.001**	-	<0.001	-

## Data Availability

The data presented in this study are available upon request from the corresponding author.

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
