# Peer review of "Genotype by Environment Interactions (G*E) of Chickens Tested in Ethiopia Using Body Weight as a Performance Trait"

_animals, 2023, doi:10.3390/ani13193121_

Round 1

Reviewer 1 Report

The research is important in supporting development of farming and farmers incomes in regions requiring extensive economic development programs. While there is major limitation to the data collection in these extensive rural communities the authors identify these in the manuscript and yet work to generate useful information above these limitations.

I have suggested a arrange of grammar and expression improvements.

Review: Genotype by environment interactions (G*E) of chickens tested in Ethiopia using body weight as performance trait.

Ln 39: remove ‘used’ add ‘was used’

Ln 39: remove ‘strains to.’ Add ‘strains’

Ln 43: remove ‘young’ add ‘limited’

Ln 44: remove ‘which are all mainly’ add ‘which, are in the main,’

Ln 49:  Remove ‘much present in African countries and was shown to contribute’ add ‘significant farming activity in African countries and contributes’

Ln 56: remove ‘of it’ add ‘of increased production’

Ln 61: remove ‘To find out beneficiary’ add ‘To determine the benefits’

Ln 66: remove ‘which are only recently applied’ add ‘which have only recently been applied’

Ln 77: remove ‘These general results of are a rough hypothesis’’ add ‘These general results are the basis of a hypothesis’

Ln 385.: remove ‘big’ add ‘large’

Ln 93: remove ‘control’ add ‘as a control’

Ln 100: remove ‘control’ add ‘that’

Ln 103; remove ‘a’

Ln 106: remove ‘production’ add ‘the production’

Ln 108: remove ‘keeping’ add ‘preference’

Ln 113: Remove ‘great’ add ‘substantial’

Ln 118: remove ‘Local Horro being control for the’ add ‘The local Horro strain was used as the control for comparison with the’

Ln 122: remove ‘approximately 34.825 birds’ add ‘approximately 35 birds’ 34.825 is more than an approximation.

Ln 123: remove ‘way’ add ‘considerably lower’

Ln 128: remove ‘farmers their’ add ‘the farmers’

Ln 129: remove ‘externally bought’ add ‘commercially sourced’

Ln 131: remove ‘supplemented feeding the’ add ‘supplementary feeding of the’

Ln152 remove ‘as this is’ add and is’

Ln 155: remove ‘having common genes,’

Ln 159: remove ‘made’ add ‘established’

Ln 160: remove ‘to cross’ add ‘in the cross’

Ln 164: remove ‘for the ability of hatching’ add ‘for their broodiness and hatching’

Ln 181: remove ‘its’

Ln 182: remove ‘warm’ add ‘warm environmental’

Ln 207: remove ‘in’

Ln 212: remove ‘al weather’ add ‘all weather’

Ln 213: remove ‘amount’ add ‘number’

Ln 217: remove ‘month’ add ‘monthly’

Ln 219: remove ‘requesting the’ add ‘as requested from the’

Ln 230: remove ‘Parameters are average’ add ‘The parametric average’

Ln 234: remove ‘requesting’

Ln 248: remove ‘weighted’ add ‘weighed’

Ln 251: remove ‘neighboring ages of weighing’ add ‘adjacent weighing ages’

Ln 252: remove ‘999 regarding females’ add ‘999 were females’  also remove ‘and regarding 989 males’ add ‘and 989 were males’

Ln 254: remove ‘free-range’ add ‘free-ranging’

Ln 255: remove ‘group weight attempt’ add ‘group weighting’

Ln 262: remove ‘are given’ add ‘and are given’

Ln 264: ’ To reduce outliers made visible by these plots and to retain observations within the range of growth mentioned in literature regarding local African or introduced chickens, outliers were removed per BW at each specific age for which the thresholds are provided in Table 4 [27,28].

Replace with: ‘These plots were used to remove outliers and retain observations within the range of growth weights mentioned in literature for the local African or introduced strains at each specific age with the thresholds provided in Table 4 [27,28].

Ln 269: remove ‘Table 5, in which the values do not vary much indicating robustness of the study.’

add ‘….Table 5. There is little variation in the values giving support the robustness of the study.

Ln 327: remove ‘as environment’ add ‘as the environment’

Ln 350 & 363: remove ‘more’

Ln 372: remove ‘stay’ add ‘remained’

Ln 378: remove ‘smaller’ add ‘lower’ also remove ‘shows to have’ add ‘had’

Ln 381: remove ‘are close together’ add ‘similar’

Ln 382: remove ‘, the performance of Koekoek and Horro in cool humid are much further 382 apart’ add ‘, the largest performance difference was between Koekoek and Horro in cool humid.’

Ln 400: remove ‘Sasso had mostly the highest predicted BW in OM, AM and TG regions, and Kuroiler 400 had mostly the highest predicted BW in AA.

Add ’ Primarily, Sasso had the highest predicted BW in OM, AM and TG regions, and Kuroiler the highest predicted BW in AA’

Ln 402: ‘can be validated by the hypothesis results in the GIS analysis study’ This lacks clarity and needs to be rephrased.

Ln 407: remove ‘absent’ add ‘in the absence of ’ also remove ‘little’ add ‘minimal’

Ln 409: remove ‘being’ add ‘having’

Ln 415 remove ‘Horro not performing well can be explained by a young breeding program established in 2008 only, giving

Add ‘Horro not performing well can be explained by the limited breeding program having only been established in 2008, having’

Ln 419: remove ‘testing’

Ln 421: remove ‘were smallest’ add ‘were minimal’

Ln 426” remove ‘on regions’ add ‘as is for regions’

Ln 443: remove ‘forgotten’ add ‘dismissed’ also remove ‘biggest’ add ‘largest’

Ln 444: remove ‘surrounding the capital its cool humid AA region’ this lacks clarity

Ln 447: remove ‘stretched’ add ‘extensive’

Ln 448: remove ‘little’ add ‘minimal’

Ln 458: remove ‘much’

Ln 462: remove ‘It are again’ add ‘Again its’

Ln 464: remove ‘bigger’ add ‘more extensive’

Ln 470: remove ‘bigger’ add ‘much more’

Ln 473: remove ‘which is additionally useful for making breeding decisions.’ Add ‘which adds useful information for making breeding decisions.’

Ln 474: remove ‘being’ add ‘are’

Ln 475: remove ‘While weather values of during the experiment’ add ‘Weather values during the experiment’

Ln 515: remove ‘pleaded’ add ‘necessary’

There are suggested improvements to the grammar provided

Author Response

Comments and Suggestions for Authors

The research is important in supporting development of farming and farmers incomes in regions requiring extensive economic development programs. While there is major limitation to the data collection in these extensive rural communities the authors identify these in the manuscript and yet work to generate useful information above these limitations.

I have suggested a arrange of grammar and expression improvements.

Review: Genotype by environment interactions (G*E) of chickens tested in Ethiopia using body weight as performance trait.

Ln 39: remove ‘used’ add ‘was used’

This is now changed as suggested by the reviewer.

Ln 39: remove ‘strains to.’ Add ‘strains’

This is now changed as suggested by the reviewer.

Ln 43: remove ‘young’ add ‘limited’

We think young is a good description of the breeding program of Horro because indeed the limitation is that the breeding program is young. If you write down limited it might not be clear for the reader what is the limitation.

Ln 44: remove ‘which are all mainly’ add ‘which, are in the main,’

This is now changed as suggested by the reviewer.

Ln 49:  Remove ‘much present in African countries and was shown to contribute’ add ‘significant farming activity in African countries and contributes’

This is now changed as suggested by the reviewer.

Ln 56: remove ‘of it’ add ‘of increased production’

The second reviewer also had a suggestion for improving this sentence and now it’s suggested according to that suggestion.

Ln 61: remove ‘To find out beneficiary’ add ‘To determine the benefits’

This is now changed as suggested by the reviewer.

Ln 66: remove ‘which are only recently applied’ add ‘which have only recently been applied’

This is now changed as suggested by the reviewer.

Ln 77: remove ‘These general results of are a rough hypothesis’’ add ‘These general results are the basis of a hypothesis’

This is now changed as suggested by the reviewer.

Ln 385.: remove ‘big’ add ‘large’

This is now changed as suggested by the reviewer. We assume the reviewer meant line 85.

Ln 93: remove ‘control’ add ‘as a control’

This is now changed as suggested by the reviewer.

Ln 100: remove ‘control’ add ‘that’

There is no ‘control’ on line 100.

Ln 103; remove ‘a’

This is now changed as suggested by the reviewer.

Ln 106: remove ‘production’ add ‘the production’

This is now changed as suggested by the reviewer.

Ln 108: remove ‘keeping’ add ‘preference’

This is now changed as suggested by the reviewer.

Ln 113: Remove ‘great’ add ‘substantial’

This is now changed as suggested by the reviewer.

Ln 118: remove ‘Local Horro being control for the’ add ‘The local Horro strain was used as the control for comparison with the’

This is now changed as suggested by the reviewer.

Ln 122: remove ‘approximately 34.825 birds’ add ‘approximately 35 birds’ 34.825 is more than an approximation.

This is now changed as suggested by the reviewer.

Ln 123: remove ‘way’ add ‘considerably lower’

This is now changed as suggested by the reviewer.

Ln 128: remove ‘farmers their’ add ‘the farmers’

This is now changed as suggested by the reviewer.

Ln 129: remove ‘externally bought’ add ‘commercially sourced’

This is now changed as suggested by the reviewer.

Ln 131: remove ‘supplemented feeding the’ add ‘supplementary feeding of the’

This is now changed as suggested by the reviewer.

Ln152 remove ‘as this is’ add and is’

We don’t think this suggestion is correct English or makes the statement clearer. Hence we left it as it was.

Ln 155: remove ‘having common genes,’

This is now changed as suggested by the reviewer.

Ln 159: remove ‘made’ add ‘established’

This is now changed as suggested by the reviewer.

Ln 160: remove ‘to cross’ add ‘in the cross’

This is now changed as suggested by the reviewer.

Ln 164: remove ‘for the ability of hatching’ add ‘for their broodiness and hatching’

This is now changed as suggested by the reviewer.

Ln 181: remove ‘its’

This is now changed as suggested by the reviewer.

Ln 182: remove ‘warm’ add ‘warm environmental’

This is now changed as suggested by the reviewer.

Ln 207: remove ‘in’

This is now changed as suggested by the reviewer.

Ln 212: remove ‘al weather’ add ‘all weather’

This is now changed as suggested by the reviewer.

Ln 213: remove ‘amount’ add ‘number’

This is now changed as suggested by the reviewer.

Ln 217: remove ‘month’ add ‘monthly’

This is now changed as suggested by the reviewer.

Ln 219: remove ‘requesting the’ add ‘as requested from the’

This is now changed as suggested by the reviewer.

Ln 230: remove ‘Parameters are average’ add ‘The parametric average’

This is now changed as suggested by the reviewer.

Ln 234: remove ‘requesting’

This is now changed as suggested by the reviewer.

Ln 248: remove ‘weighted’ add ‘weighed’

This is now changed as suggested by the reviewer.

Ln 251: remove ‘neighboring ages of weighing’ add ‘adjacent weighing ages’

This is now changed as suggested by the reviewer.

Ln 252: remove ‘999 regarding females’ add ‘999 were females’  also remove ‘and regarding 989 males’ add ‘and 989 were males’

This is now changed as suggested by the reviewer.

Ln 254: remove ‘free-range’ add ‘free-ranging’

This is now changed as suggested by the reviewer.

Ln 255: remove ‘group weight attempt’ add ‘group weighting’

This is now changed as suggested by the reviewer.

Ln 262: remove ‘are given’ add ‘and are given’

This is now changed as suggested by the reviewer. But also ‘which’ is removed in this sentence to make it better floating according to us.

Ln 264: ’ To reduce outliers made visible by these plots and to retain observations within the range of growth mentioned in literature regarding local African or introduced chickens, outliers were removed per BW at each specific age for which the thresholds are provided in Table 4 [27,28].

Replace with: ‘These plots were used to remove outliers and retain observations within the range of growth weights mentioned in literature for the local African or introduced strains at each specific age with the thresholds provided in Table 4 [27,28].

This is now changed as suggested by the reviewer.

Ln 269: remove ‘Table 5, in which the values do not vary much indicating robustness of the study.’

add ‘….Table 5. There is little variation in the values giving support the robustness of the study.

This is now changed as suggested by the reviewer.

Ln 327: remove ‘as environment’ add ‘as the environment’

This is now changed as suggested by the reviewer.

Ln 350 & 363: remove ‘more’

This is now changed as suggested by the reviewer for line 350. On line 363 there is no ‘more’.

Ln 372: remove ‘stay’ add ‘remained’

This is now changed as suggested by the reviewer.

Ln 378: remove ‘smaller’ add ‘lower’ also remove ‘shows to have’ add ‘had’

This is now changed as suggested by the reviewer.

Ln 381: remove ‘are close together’ add ‘similar’

This is now changed as suggested by the reviewer.

Ln 382: remove ‘, the performance of Koekoek and Horro in cool humid are much further 382 apart’ add ‘, the largest performance difference was between Koekoek and Horro in cool humid.’

This is now changed as suggested by the reviewer.

Ln 400: remove ‘Sasso had mostly the highest predicted BW in OM, AM and TG regions, and Kuroiler 400 had mostly the highest predicted BW in AA.

Add ’ Primarily, Sasso had the highest predicted BW in OM, AM and TG regions, and Kuroiler the highest predicted BW in AA’

This is now changed as suggested by the reviewer.

Ln 402: ‘can be validated by the hypothesis results in the GIS analysis study’ This lacks clarity and needs to be rephrased.

This is now added to the sentence: ‘since they used almost the same dataset but a very different analysis approach.’

Ln 407: remove ‘absent’ add ‘in the absence of ’ also remove ‘little’ add ‘minimal’

This is now changed as suggested by the reviewer.

Ln 409: remove ‘being’ add ‘having’

This is now changed as suggested by the reviewer.

Ln 415 remove ‘Horro not performing well can be explained by a young breeding program established in 2008 only, giving

Add ‘Horro not performing well can be explained by the limited breeding program having only been established in 2008, having’

This is now changed as suggested by the reviewer.

Ln 419: remove ‘testing’

This is now changed as suggested by the reviewer.

Ln 421: remove ‘were smallest’ add ‘were minimal’

This is now changed as suggested by the reviewer.

Ln 426” remove ‘on regions’ add ‘as is for regions’

This is now changed as suggested by the reviewer.

Ln 443: remove ‘forgotten’ add ‘dismissed’ also remove ‘biggest’ add ‘largest’

This is now changed as suggested by the reviewer.

Ln 444: remove ‘surrounding the capital its cool humid AA region’ this lacks clarity

We think explaining that region OM surrounds AA is important. To make the sentence clearer it has been changed to: ‘OM surrounds the capital which is the cool humid AA region.’

Ln 447: remove ‘stretched’ add ‘extensive’

This is now changed as suggested by the reviewer.

Ln 448: remove ‘little’ add ‘minimal’

This is now changed as suggested by the reviewer.

Ln 458: remove ‘much’

This is now changed as suggested by the reviewer. We assume it is supposed to be the ‘much’ at line 451.

Ln 462: remove ‘It are again’ add ‘Again its’

This is now changed as suggested by the reviewer. Although we used ‘it is’ instead of ‘its’.

Ln 464: remove ‘bigger’ add ‘more extensive’

This is now changed as suggested by the reviewer.

Ln 470: remove ‘bigger’ add ‘much more’

There is no ‘bigger’ at line 470.

Ln 473: remove ‘which is additionally useful for making breeding decisions.’ Add ‘which adds useful information for making breeding decisions.’

This is now changed as suggested by the reviewer.

Ln 474: remove ‘being’ add ‘are’

This is now changed as suggested by the reviewer.

Ln 475: remove ‘While weather values of during the experiment’ add ‘Weather values during the experiment’

This is now changed as suggested by the reviewer.

Ln 515: remove ‘pleaded’ add ‘necessary’

The second reviewer also had a suggestion for improving this sentence and now it’s suggested according to that suggestion.

Comments on the Quality of English Language

There are suggested improvements to the grammar provided.

These previous comments are all addressed now.

Reviewer 2 Report

I have provided my comments and suggestions in the attached review report.

The quality of English Language is generally satisfactory. I have made suggestions to enable the authors correct areas that need improvement.

Author Response

[Animals] Manuscript ID: animals-2621020 - Review Report

Genotype-environment interactions are important when new genotypes are introduced into novel environments and therefore the study will help to properly match tropically improved adapted chicken strains to various environments in the study area. The background of the study, objectives and experimental procedures employed are satisfactory and easy to appreciate. However, authors are invited to improve on the discussion and conclusion by adhering to the suggestions indicated in Table 1. Authors need to improve on the conclusions and isolate clear recommendations for industry and further research. The manuscript is generally well written and my comments in Table 1 are to help the authors to improve on the quality of the manuscript as a guide for poultry improvement efforts and future research in Africa.

Table 1. Comments and Suggestions for Authors

Line(s)     Comment/Suggestion/Correction requested

2-3         A suggestion for modification of the title has been provided.

We couldn’t find the suggestion for the modification in the reviewer their letter. Hence we leave the title as it is.

18          Edit “Poultry and smallholder farming” to read “Smallholder poultry farming”.

This is now changed as suggested by the reviewer.

19          Edit “livelihoods” to read “households”.

This is now changed as suggested by the reviewer.

25          Edit “relative performances are different” to read “these differ”.

This is now changed as suggested by the reviewer.

46          Avoid using words in the title as keywords.

This is not mentioned as a guideline for keywords:

https://www.mdpi.com/journal/animals/instructions : ‘Three to ten pertinent keywords need to be added after the abstract. We recommend that the keywords are specific to the article, yet reasonably common within the subject discipline.’

Since we think our keywords are specific for the article we just leave them as they are.

49          Edit “much present in” to read “prevalent in most”.

The first reviewer also had a suggestion for improving this sentence and now it’s suggested according to that suggestion.

55-56       Edit “important for flexible implementation of it” to read “thus needed”.

This is now changed as suggested by the reviewer.

66          Edit “which are only recently applied in livestock research” to read “a novel methodology               in livestock”.

The first reviewer also had a suggestion for improving this sentence and now it’s suggested according to that suggestion.

90-91       Edit “is conducted with pure breed as well as” to read “was on purebred and”.

This is now changed as suggested by the reviewer.

98          Authors should ensure consistency by sticking to “S*E” from this portion.

If we refer to results with our data we always stick to calling it S*E. However, in this sentence we refer to classic G*E as a analysis methodology. This does not refer to specifically to our data results with strains as genotypes. Hence saying ‘classic G*E’ is appropriate here.

100         Correct “This, as opposed” to read “This is in contrast”

This is now changed as suggested by the reviewer.

103-104    Reword sentence for clarity.

The first reviewer also had a suggestion for improving this sentence and now it’s suggested according to that suggestion.

141         In justifying the lack of animal ethics approval for the experiment, authors should provide reference(s) in support of the “ARRIVE Guidelines”.

This reference is now added.

142-190    Some images of the different strains used in the study will be useful to readers.

These images are provided in the graphical abstract.

303         Appropriate reference(s) needed to support the “ANOVA” method.

We believe ANOVA is a commonly used method in quantitative genetics and other sciences using statistical approaches on data. Therefore we expect that readers of the manuscript will be familiar with it or a simple google search will reveal it for them. The program we used for the statistical analysis is clearly cited: ‘A linear fixed effects model was implemented, using PROC GLM, SAS version 9.4 [30].’

391-392;

400 and 420 Authors should explain all acronyms used in the “Discussion” section in the first instance of use.

This is not needed according to the instructions for Authors: https://www.mdpi.com/journal/animals/instructions :

Acronyms/Abbreviations/Initialisms should be defined the first time they appear in each of three sections: the abstract; the main text; the first figure or table. When defined for the first time, the acronym/abbreviation/initialism should be added in parentheses after the written-out form.’

405         Edit “Bear in mind” to read “It is worthy of note”.

This is now changed as suggested by the reviewer.

415         Authors should reword “young breeding programme” for clarity.

The first reviewer also had a suggestion for improving this sentence and now it’s suggested according to that suggestion.

418         Delete “as”.

This is now changed as suggested by the reviewer.

419         Authors should reword “tested on-farm testing” for clarity.

The first reviewer also had a suggestion for improving this sentence and now it’s suggested according to that suggestion.

421-422    Authors should reword “with lacking clear explanations for this performance similarity” for clarity.

It is re-worded now to: ‘Clear explanations for this performance similarity are lacking.’

428         Authors should reword “Sasso or Kuroiler having highest predicted BW” for clarity.

We don’t think this sentence need rephrasing. We think the statement is clear as it is and reflects back on the previous paragraphs in the story.

432         Correct “outperforming” to read “outperformed”.

This is now changed as suggested by the reviewer. The word ‘that’ is also added in the sentence to make the structure more correct.

437         Correct “strain are of interest for” to read “strains could be subjects of”.

2

This is now changed as suggested by the reviewer.

462-463     Authors should reword “It are again precipitation variables” for clarity.

The first reviewer also had a suggestion for improving this sentence and now it’s suggested according to that suggestion.

467-468      Reference(s) needed to support “AM, just as OM and TG are all part of the cool

sub humid AEZ”.

The reference is now provided: ‘AM, just as OM and TG are all part of the cool sub humid AEZ (Table 1),’

515          Correct “pleaded” to read “recommended”.

This is now changed as suggested by the reviewer.

517-536     Authors need to improve on the conclusions and isolate clear recommendations

for industry and further research. Conclusions are summarized findings of the

study based on the set objectives.

It is not specifically mentioned how the conclusion should be structured or what information it should contain according to the journal: https://www.mdpi.com/journal/animals/instructions :

Conclusions: This section is mandatory, with one or two paragraphs to end the main text.’

We do agree with the suggestion of this reviewer that the conclusion should contain some recommendations for further research. Hence the following sentence is added:

‘The findings of this study are a good indication for breed preferences based on BW in certain Ethiopian regions or agro-ecologies. Outcomes could be used by local farmers who keep chickens as well as breeders who want to further improve certain breeds for various conditions. The on-farm data collection of this study generates a unique, highly valuable, but also limited dataset. Future research could try to overcome these limitations by either upscaling the sample size, which will be highly labor intensive, or by conducting on research station data collection. However, the latter is expected to result in G*E presence, in which chickens have higher performances on-station compared to if the chickens were kept on-farm, while this on-farm environment is where chickens eventually have to be productive in.’

526          Correct “Findings” to read “The current findings”.

This is now changed as suggested by the reviewer.

528-530     These read more like a repetition of portions of the discussion.

It is not specifically mentioned how the conclusion should be structured or what information it should contain according to the journal: https://www.mdpi.com/journal/animals/instructions :

Conclusions: This section is mandatory, with one or two paragraphs to end the main text.’

Since we believe mentioning the most important results and discussion points of the manuscript for the reader, so he/she understands the key points of the manuscript, we believe this sentence is appropriate.

Comments on the Quality of English Language

The quality of English Language is generally satisfactory. I have made suggestions to enable the authors correct areas that need improvement.

This is now improved according to the suggestions of the reviewer.
